# Impacts of Managed Vegetation Restoration on Arbuscular Mycorrhizal Fungi and Diazotrophs in Karst Ecosystems

**DOI:** 10.3390/jof10040280

**Published:** 2024-04-10

**Authors:** Mingming Sun, Dan Xiao, Wei Zhang, Kelin Wang

**Affiliations:** 1Key Laboratory of Agro-Ecological Processes in Subtropical Region, Institute of Subtropical Agriculture, Chinese Academy of Sciences, Changsha 410125, China; smm23@mails.ucas.ac.cn (M.S.); zhangw@isa.ac.cn (W.Z.); 2Huanjiang Agriculture Ecosystem Observation and Research Station of Guangxi, Guangxi Key Laboratory of Karst Ecological Processes and Services, Huanjiang 547100, China; 3University of Chinese Academy of Sciences, Beijing 100039, China; 4Huanjiang Observation and Research Station for Karst Ecosystems, Chinese Academy of Sciences, Huanjiang 547100, China

**Keywords:** arbuscular mycorrhizal fungi, diazotroph, fragile ecosystems, plantation restoration

## Abstract

The crucial functional arbuscular mycorrhizal fungi (AMF) and diazotrophs play pivotal roles in nutrient cycling during vegetation restoration. However, the impact of managed vegetation restoration strategies on AMF and diazotroph communities remains unclear. In this study, we investigated the community structure and diversity of AMF and diazotrophs in a karst region undergoing managed vegetation restoration from cropland. Soil samples were collected from soils under three vegetation restoration strategies, plantation forest (PF), forage grass (FG), and a mixture of plantation forest and forage grass (FF), along with a control for cropland rotation (CR). The diversity of both AMF and diazotrophs was impacted by managed vegetation restoration. Specifically, the AMF Shannon index was higher in CR and PF compared to FF. Conversely, diazotroph richness was lower in CR, PF, and FG than in FF. Furthermore, both AMF and diazotroph community compositions differed between CR and FF. The relative abundance of AMF taxa, such as *Glomus*, was lower in FF compared to the other three land-use types, while *Racocetra* showed the opposite trend. Among diazotroph taxa, the relative abundance of *Anabaena*, *Nostoc*, and *Rhizobium* was higher in FF than in CR. Soil properties such as total potassium, available potassium, pH, and total nitrogen were identified as the main factors influencing AMF and diazotroph diversity. These findings suggest that AMF and diazotroph communities were more sensitive to FF rather than PF and FG after managed vegetation restoration from cropland, despite similar levels of soil nutrients among PF, FG, and FF. Consequently, the integration of diverse economic tree species and forage grasses in mixed plantations notably altered the diversity and species composition of AMF and diazotrophs, primarily through the promotion of biocrust formation and root establishment.

## 1. Introduction

Soil key functional microorganisms serve as vital indicators for assessing soil quality and maintaining soil fertility. AMF are known to effectively improve nutrient utilization efficiency and regulate the stability of plant community ecosystems [1,2,3]. The mutualistic relationship between AMF and plants is particularly significant during vegetation restoration, considering that approximately 80% of plant species engage in mycorrhizal associations [3,4]. Additionally, free-living nitrogen (N) fixation driven by diazotrophs is an essential source of N input, particularly in the absence of symbiotic N fixation by legumes [5,6]. Therefore, comprehending how variations in the community composition, diversity, and abundance of AMF and diazotrophs respond to different vegetation restoration strategies can offer valuable insights into promoting positive vegetation succession and ensuring ecosystem stability [7].

The communities of AMF and diazotrophs are influenced by a range of biotic and abiotic factors [3,8,9]. Vegetation restoration indirectly impacts AMF by altering soil nutrient availability and plant communities, a relationship that has been widely studied [7,10,11]. These studies consistently reveal a significant correlation between the structure and function of AMF and diazotroph communities and plant diversity and soil quality. Plant diversity can have a positive, negative, or neutral effect on AMF and diazotroph diversity. These varying impacts may be attributed to a myriad of factors, including the specific plant species involved, the type of soil, and the duration of vegetation recovery [3,12,13,14,15]. Compared to agricultural farming systems, vegetation cover resulting from restoration efforts affects AMF and diazotroph diversity and populations by influencing litter input and root exudates [7,16]. Increased availability of soil nutrients, such as carbon (C) sources, promotes microbial growth, leading to higher abundances of AMF and diazotrophs [4,6,16]. Conversely, excessive N fertilizer input has been shown to negatively impact diazotroph abundance by suppressing N_2_ fixation activity [17,18]. Furthermore, tillage practices can disrupt hyphae, resulting in lower AMF colonization in croplands [19,20]. Therefore, the effects of vegetation restoration on AMF and diazotroph communities are complex due to differences in soil properties, plant characteristics, and management practices. Understanding the response of AMF and diazotroph diversity and community composition to vegetation restoration, especially in managed systems, is crucial for effective land management.

The southwestern region of China boasts the largest karst landscape in the world, an environmentally fragile terrain characterized by shallow soil layers, limited soil nutrients, and a unique dual structure of aboveground and underground features [21,22]. Over the past century, the area has experienced exacerbated rock desertification due to intense tilling activities. In response, various ecological restoration initiatives, such as the “Grain to Green” project, have been implemented [21,23]. A key aspect of restoration efforts lies in managed vegetation restoration, encompassing the strategic planting of economic tree species and forage grasses to restore degraded land in karst areas [24,25,26]. Due to the accelerated depletion of soil nutrients caused by unreasonable land-use in the early stages, vegetation restoration efforts in this area were frequently constrained by soil nutrient deficiencies, particularly N [27]. As a result, enhancing the functional capacity of key microorganisms, such as AMF and diazotrophs, in the vegetation restoration process significantly contributes to the improvement in soil nutrient levels.

Many studies have reported on the impact of vegetation restoration on soil nutrient accumulation and microbial communities [7,16,25,26,28]. However, the majority of these investigations have focused on natural vegetation restoration, with limited research conducted on managed vegetation restoration. Previous studies conducted in the karst region have shown that planting economic tree species and elephant grass, compared to cropland, leads to increased C and N stocks and enhances the abundance of total phospholipid fatty acids [24,26]. Moreover, managed vegetation restoration has been found to alleviate microbial C limitation but exacerbate limitations in microbial N and phosphorus (P) [28]. These findings indicate that managed restoration strategies exert a significant influence on soil nutrient levels and microbial abundance. Nevertheless, despite the recognized significance of AMF and diazotrophs in managed vegetation restoration efforts, the impact of plantations with forage grass and economic tree species on these microbial communities remains unclear to date.

To gain a deeper understanding of the changes in AMF and diazotroph characteristics across various managed vegetation restoration strategies, this study aims to assess the community composition and diversity of AMF and diazotrophs in soils under three strategies. We hypothesized that managed vegetation restoration, compared to cropland, would increase the diversity of both AMF and diazotrophs and alter the community composition as well.

## 2. Materials and Methods

### 2.1. Study Area

The study area is situated within the Guzhou catchment (107°56′–107°57′ E and 24°54′ E–24°55′ N) in Huanjiang County, Guangxi Zhuang Autonomous Region, Southwest China (Figure 1). The climate of this region is characterized as subtropical monsoon, experiencing an average annual precipitation of 1750 mm and an average annual temperature of 19.6 °C. Based on the FAO/UNESCO soil classification system, the soil prevalent in this region is categorized as calcareous lithosol, which has developed from limestone.

During the 1980s, a significant intensification of agricultural activities resulted in a pronounced rocky desertification of the landscape. However, over the past two decades, the “Grain for Green” project has been instrumental in promoting vegetation restoration through strategic vegetation management in the area, including in the downhill position. Four land-use types were chosen for assessment, including corn–soybean cropland (CR) as a control, and three managed vegetation restoration strategies were incorporated: plantation forest (PF, highlighted by *Zenia insignis*), forage grass (FG, utilizing Guimu-1 hybrid elephant grass), and a mixture of plantation forest and forage grass (FF, integrating both *Zenia insignis* and Guimu-1 hybrid elephant grass). These latter three management approaches were implemented over a duration of approximately 20 years. All sample sites were chosen to have identical geochemical backgrounds, soil types, locations (downhill position), slopes (approximately 20°), and slope directions (southeast), ensuring consistency across all sample sites.

### 2.2. Soil Sampling

Field sampling was conducted in August 2022. For each type of site, four representative sites were selected. Before sampling, the aboveground humus layer was removed. At each sampling point, eight soil cores were extracted from the upper 0–15 cm layer and then combined to create a composite sample. A total of 16 samples were collected and sent to the laboratory, where they were filtered through a 2 mm sieve to remove stones, plant residues, and animal carcasses. Each sample was then divided into two subsamples. One subsample was air-dried for the determination of soil physico-chemical properties, while the other subsample was stored at −80 °C for the detection of AMF and diazotroph communities.

### 2.3. Soil Analysis

Soil organic carbon (SOC) was quantified using the dichromate redox colorimetric method. This involved the use of a mixture of KCr_2_O_7_ and H_2_SO_4_, followed by titration with FeSO_4_ to determine the carbon content. Total nitrogen (TN) content was determined using an elemental analyzer (EA 3000; EuroVector, Milano, Italy). Soil ammonium nitrogen (NH_4_^+^) and nitrate nitrogen (NO_3_^−^) were analyzed using an autoanalyzer after KCl extraction procedures. Total phosphorus (TP) was quantified using the ascorbic acid molybdate method, while available phosphorus (AP) was determined via the molybdenum blue method. Total potassium (TK) and available potassium (AK) were analyzed using flame photometry after extraction with sodium hydroxide and ammonium acetate, respectively. Exchangeable calcium (Ca^2+^) and magnesium (Mg^2+^) were extracted through a compulsory exchange in ammonium acetate at pH 7 and subsequently analyzed by inductively coupled plasma atomic emission spectrometry (ICP-AES). Microbial carbon (MBC) and microbial nitrogen (MBN) were determined using the chloroform fumigation and extraction method. Soil dissolved organic carbon (DOC) was extracted with 0.5 M K2SO4 (*w*/*v* of 1/5) and then determined using a total organic C analyzer. Soil-dissolved organic nitrogen (DON) was calculated as the difference between total dissolved nitrogen and mineral nitrogen (NH_4_^+^ and NO_3_^−^). Soil pH values were measured using a pH meter (FE20K; Mettler-Toledo, Greifensee, Switzerland), employing a 2.5:1 water-to-soil ratio. The comprehensive descriptions of all soil properties have been provided in our previous studies [6,7,26].

### 2.4. DNA Extraction and Amplicon Sequencing

DNA was extracted from 0.5 g soil samples using the FastDNA Spin kit for soil (MP Biomedicals, Santa Ana, CA, USA). The 18S rRNA gene, specific to soil AMF, was amplified through a nested polymerase chain reaction (PCR) approach. This involved the use of primers AML1 (ATCAACTTTCGATGGTAGGATAGA) and AML2 (GAACCCAAACACTTT-GGTTTCC) [29], as well as AMV4.5NF (AAGCTCGTAGTT-GAATTTCG) and AMDGR (CCCAACTATCCCTATTAATCAT) [30] for the first and second rounds of amplification, respectively. The PCR reaction mixture comprised 10 μL of 2 × PCR ExTaq, 1 μL of a DNA template, and 0.50 μL of each primer, with sterile water added to achieve the desired final concentrations. The DNA template for the second round of PCR amplification was diluted 50-fold using the product obtained from the first-round PCR amplification. Additionally, a primer set comprising nifH-F (AAAGGYGGWATCGGYAARTCCACCAC) and nifH-R (TTGTTSGCSGCRTACATSGCCATCAT) was employed to amplify the *nifH* gene for diazotrophs. Further details regarding the PCR conditions were presented in our previous studies [6,31]. The purification of the PCR amplicons for AMF and the diazotroph *nifH* gene was performed prior to sequencing. Subsequently, sequencing was conducted on the Illumina MiSeq PE300 platform at Shanghai Majorbio Bio-Pharm Technology Co., Ltd. located in Shanghai, China.

### 2.5. Sequence Analysis

The AMF and *nifH* gene sequences were processed using QIIME v2-2020.2. For the *nifH* gene, sequences of pseudogenes and homologous genes were removed using the NifH Miseq Illumina Amplicon Analysis Pipeline (NifMAP, https://github.com/roey-angel/NifMAP, accessed on 17 February 2024). Raw reads of both genes were initially imported into QIIME and processed to generate amplicon sequence variants (ASVs) using the “q2-DATA2” plugin (https://github.com/qiime2/q2-dada2, accessed on 18 February 2024). NifMAP was then applied to filter sequences of nifH ASVs for pseudogenes and homologs. The classification of AMF and *nifH* ASVs was performed using the “q2-feature-classifier” plugin (https://github.com/qiime2/q2-feature-classifier, accessed on 18 February 2024), in conjunction with the PR2 v4.13.0 and *nifH* reference databases (https://blogs.cornell.edu/buckley/nifh-sequence-database/, accessed on 18 February 2024). To ensure taxonomic accuracy, all taxa were verified against the NCBI taxonomy database. The richness and Shannon index for AMF and diazotroph *nifH* genes were calculated by normalizing the ASV abundance table using the R package “RAM” v1.2.1.7.

### 2.6. Statistical Analysis

The data were initially verified for normal distribution, and transformation was applied as necessary. The Duncan’s test (*p* < 0.05) was used to assess significant differences in soil properties, the diversity of AMF (arbuscular mycorrhizal fungi) and diazotrophs, and the relative abundance of their taxa across different land-use types. Non-metric multidimensional scaling (NMDS) was employed to evaluate the distribution of AMF and diazotroph community composition across different vegetation restoration strategies. A Pearson correlation analysis was conducted to examine the relationships between the genera of AMF and diazotrophs, elucidating potential associations between these microbial groups. Random forest models were employed to determine the relative importance of soil properties for predicting the abundance or diversity of AMF and diazotrophs. All statistical analyses were performed using R version 4.02.

## 3. Results 

### 3.1. Soil Properties and the Diversity of AMF and Diazotrophs

Soil properties were influenced by managed vegetation restoration practices. When comparing the effects of vegetation restoration in PF, FG, and FF, it was observed that the concentrations of Mg^2+^, NO_3_^−^, TP, AP, and AK were significantly higher in the control of CR. Conversely, the NH_4_^+^ content was lower in the CR compared to PF, FG, and FF. Additionally, the MBC and DON were found to be highest in the PF, exceeding those in the CR, FG, and FG (Figure 2). 

In this study, vegetation restoration from cropland exhibited a significant effect on the Shannon index of AMF, as well as the richness of diazotrophs. Specifically, the Shannon index of AMF in the CR and PF treatments was significantly higher than in the FF treatment. Additionally, diazotroph richness was found to be higher in the FF treatment than in CR, PF, and FG (Figure 3).

### 3.2. The community Compositions of AMF and Diazotrophs

Seven primary AMF genera were identified with a relative abundance exceeding 1%, including *Acaulospora*, *Claroideoglomus*, *Diversispora*, *Glomus*, *Paraglomus*, *Racocetra*, and *Septoglomus*. Among them, *Glomus* (32.3–91.6%), *Claroideoglomus* (1.8–19.2%), and *Acaulospora* (2.3–4.6%) were determined as the dominant taxa at the genus level (Figure 4a). Regarding diazotrophs, seven genera were identified with a relative abundance of at least 1%, including *Anabaena*, *Azospirillum*, *Azotobacter*, *Bradyrhizobium*, *Frankia*, *Nostoc*, and *Rhizobium*. Notably, *Bradyrhizobium* (76.7–91.7%) and *Azotobacter* (0.6–1.7%) exhibited the highest relative abundances across all land-use types (Figure 4b).

The community compositions of AMF and diazotrophs varied significantly with managed vegetation restoration from cropland, as evidenced by the NMDS analysis. Specifically, distinct differences in AMF community composition were observed among CR, FG, and FF treatments (Figure 4c). The relative abundance of *Glomus* was higher in CR, PF, and FG compared to FF. Conversely, the relative abundance of *Racocetra* was lower in CR, PF, and FG than in FF (Figure 5). Similarly, the diazotroph community composition exhibited differences between CR and FF treatments (Figure 4d). The relative abundance of *Nostoc* and *Rhizobium* was higher in FF than in CR. Additionally, the relative abundance of *Anabaena* was higher in FF compared to CR, PF, and FG (Figure 6).

We conducted a further analysis to investigate the relationship between AMF and diazotroph taxa at the genus level. The relative abundance of AMF taxa *Acaulospora* was found to be negatively correlated with the diazotroph taxa *Nostoc*. Additionally, the relative abundance of AMF taxa *Glomus* exhibited a negative correlation with the diazotroph taxa *Anabaena*. Furthermore, the relative abundance of AMF taxa *Racocetra* showed a positive correlation with diazotroph taxa groups including *Anabaena*, *Azotobacter*, and *Nostoc* (Figure 7).

### 3.3. Drivers of AMF and Diazotroph Abundance and Diversity

In this study, random forest models were employed to identify the primary factors driving variations in AMF and diazotroph communities. It was determined that changes in AMF richness were predominantly associated with TK, pH, and SOC (Figure 8a). Moreover, the variations observed in the Shannon index of AMF were mainly predicted by TK, pH, and TN (Figure 8b). Additionally, DON, TK, and TN were the main factors predicting variations in diazotroph richness (Figure 8c). Furthermore, AK, TN, and pH were identified as the primary factors controlling changes in the diazotroph Shannon index (Figure 8d).

## 4. Discussion

This study investigated the community structure and diversity of AMF and diazotrophs under three managed vegetation restoration strategies. Our findings suggest that those managed restoration strategies, especially the FF approach, have a significant impact on the diversity and community composition of both AMF and diazotrophs. Importantly, this research highlights the distinct responses of key microbial groups to different restoration strategies. This suggests that the specific functions of those microbes within each restoration context should be carefully considered when designing and implementing future vegetation restoration efforts.

### 4.1. The Effect of Managed Vegetation Restoration on AMF and Diazotroph Diversity 

Many studies have underscored the importance of changes in soil properties and plant communities in shaping the diversity of AMF and diazotrophs during vegetation restoration [7,10,32,33]. The findings of our study demonstrate that diazotroph and AMF diversity exhibits distinct responses to managed vegetation restoration. Notably, both AMF and diazotroph diversities are found to be more sensitive to the FF strategy than the PF and FG strategies. A previous study reported higher biomass of biocrusts in the FF restoration strategy compared to other land-use types [26]. The increased formation and development of biocrusts in the FF may contribute to promoting biological N fixation [26,34,35]. Another study in the same area found higher root biomass associated with planting forage grass compared to economic tree species, thereby increasing C resource availability in the soil under forage grass planting conditions [24]. Therefore, the increased N fixation activity and C exudation observed in the FF strategy could potentially stimulate a greater diversity of diazotroph species, leading to a higher richness of diazotrophs in the FF compared to CR, PF, and FG strategies.

Contrary to our initial hypothesis, the AMF Shannon index was lower in the FF treatment compared to CR and PF. Several studies have shown that AMF diversity is correlated with plant diversity [4,36]. In the present study, economic tree species were planted in the PF treatment on downhill positions for 25 years. The presence of various grass and shrub species growing under these trees in the PF led to higher plant diversity compared to the forage grass treatments in FG and FF. Therefore, the increased plant species diversity with grasses and shrubs under PF may contribute to higher AMF diversity in PF than in FF. Additionally, a previous study found an increased saturated water adsorption ratio of biocrust in the combined restoration strategy of FF [26]. This suggests that more AMF species may be induced in the drier conditions of CR and PF compared to FF. Moreover, the combined restoration strategy involving plantation trees and forage grassland may reduce light availability due to shading, potentially suppressing AMF colonization and subsequently decreasing AMF diversity in FF [37,38]. Those findings indicate that the different changes in AMF and diazotroph diversity to managed vegetation restoration are influenced by soil condition and plant properties. Consequently, it is crucial to consider these factors when designing and implementing vegetation restoration strategies in order to maximize microbial diversity and enhance ecosystem functionality.

### 4.2. Effect of Managed Vegetation Restoration on AMF and Diazotroph Community Compositions

Consistent with our previous research in karst regions for agricultural or natural ecosystems, the genera *Glomus* and *Bradyrhizobium* emerged as the dominant genera for AMF and diazotrophs, respectively [15,16]. That suggests that *Glomus* and *Bradyrhizobium* play important roles in improving nutrient accumulation during managed vegetation restoration. Both AMF and diazotroph community compositions differed significantly between CR and FF. *Glomus* typically exhibits a wide range of adaptability to various environmental conditions, including harsh soil conditions [39,40,41]. The relative abundance of *Glomus* was lower in the FF compared to CR, PF, and FG, indicating that FF with low light availability may suppress the growth of *Glomus*. Conversely, the relative abundance of *Racocetra* was higher in FF compared to other land-use types, indicating that mixed plantation restoration strategies may influence the distribution of AMF taxa. Regarding diazotroph taxa, FF exhibited a higher relative abundance of *Anabaena*, *Nostoc*, and *Rhizobium* compared to CR. As mentioned previously, FF favors the establishment and development of biological crusts, leading to enhanced C exudation [26]. That explains the abundant presence of N-fixing *Nostoc* and *Rhizobium* in FF, supported by other studies that suggest biological crusts may result in high proportions of N-fixing Nostocales under grassland degradation in the Tibetan Plateau [42]. In addition, low soil nutrient (e.g., NO_3_^−^, AP, and AK) conditions may serve as the primary stimulus for the proliferation of cyanobacteria genera like *Anabaena* and *Nostoc*. These cyanobacteria have been observed to release nutrients, particularly N, thereby providing additional nutrients for plants. Additionally, *Nostoc* has been noted to thrive in barren soils, indicating its resilience in nutrient-poor environments [43,44].

We further investigate the relationship between AMF and diazotroph taxa in order to predict and assess nutrient cycling within the soil. The highest relative abundance of *Glomus* was negatively correlated with diazotroph groups containing *Anabaena* and other low-relative-abundance taxa, suggesting a competitive relationship between *Glomus* and *Anabaena*. In contrast, AMF taxa such as *Racocetra* and unclassified taxa were positively correlated with diazotroph groups containing *Anabaena* and *Nostoc*. A previous study found that strengthening mutualistic associations among certain diazotrophs (e.g., *Bradyrhizobium* and *Azotobacter*) and AMF (e.g., *Racocetra*) groups mainly contributes to free-living N fixation [6]. Consequently, N-fixing *Anabaena* and *Nostoc* may rely on *Racocetra* to enhance N fixation during managed vegetation restoration efforts.

### 4.3. Implications for Future Managed Vegetation Restoration

Consistent with previous studies, managed vegetation restoration increased soil C and N accumulation [24,26,28]. The NH_4_^+^ was higher in the PF, FG, and FF than in the CR because managed vegetation restoration biocrusts promoted N fixation but decreased nitrification. One study also found the positive correlation between N availability and diazotrophs [31], which agrees with the findings of this study that soil N content (e.g., DON and TN) showed significant contributions to various diazotroph diversities. Conversely, higher TP and AP contents were observed in the CR than in other restoration strategies. This suggests that excessive P availability may have an inhibitory effect on nitrogen-fixing microorganisms. Previous studies have shown that soil P levels are negatively correlated with N-fixing microorganisms and plant growth, as high P levels can limit plant growth and N assimilation [45]. Additionally, soil properties (e.g., NH_4_^+^, TP, and AP) were similar in the FF compared to PF and FG, likely due to the relatively short duration (20 years) of the different restoration strategies. However, only FF, but not PF and FG, had an effect on the diversity and certain taxa of AMF and diazotrophs, possibly because the combination of forest and grass in the FF could benefit the formation of biocrusts [26]. As aforementioned, biocrusts in FF may enhance biological N fixation by stimulating diazotrophs. Furthermore, variations in AMF diversity were primarily explained by TK, soil pH, SOC, and TN. TK is an essential nutrient for promoting plant growth, which indirectly affects AMF diversity during managed vegetation restoration [46,47].

In summary, our study demonstrates that the FF strategy offers advantages in terms of improving diazotroph diversity while also resulting in a decline in AMF diversity. This is likely due to the rich biocrust biomass, root biomass, and C exudates present in the FF strategy. Future efforts to restore ecological balance in vulnerable karst areas may benefit from considering the inoculation of AMF and diazotrophs in replanting strategies.

## 5. Conclusions

This study expands the understanding of the effects of managed vegetation restoration in vulnerable karst areas on AMF and diazotroph diversity, as well as community composition. Our study revealed that the diversity and community composition of AMF and diazotrophs were more sensitive to the FF strategy compared to PF and FG. The combination restoration strategy (i.e., FF) may promote N accumulation by increasing diazotroph diversity and strengthening the correlation among certain AMF taxa (e.g., *Racocetra*) and diazotroph taxa (e.g., *Anabaena* and *Nostoc*), although AMF diversity could be suppressed in the FF. The findings of our study provide valuable new insights for evaluating ecological restoration efforts in fragile karst regions. Given the uncertainty surrounding the long-term effects of various replantation strategies on AMF and diazotroph communities, it is crucial to continue to assess these microbial populations as a key aspect of karst vegetation restoration efforts.

## Figures and Tables

**Figure 1 jof-10-00280-f001:**
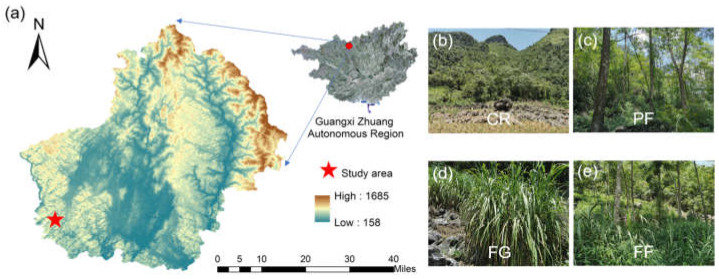
(**a**) Schematic map showing the location of the study area and the selection of four land-use types, including (**b**) Cropland rotation (CR); (**c**) plantation forest (PF); (**d**) forage grass (FG); and (**e**) plantation forest and forage grass mixed (FF).

**Figure 2 jof-10-00280-f002:**
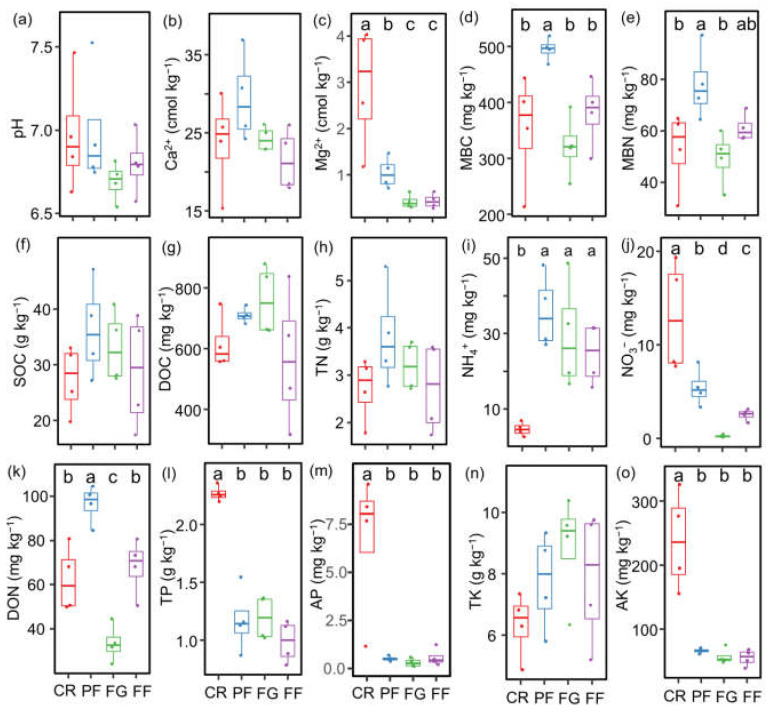
Change in soil properties during vegetation restoration from cropland. (**a**) Soil pH (pH); (**b**) soil exchangeable Ca^2+^ (Ca^2+^); (**c**) soil exchangeable Mg^2+^ (Mg^2+^); (**d**) microbial biomass carbon (MBC); (**e**) microbial biomass nitrogen (MBN); (**f**) soil organic carbon (SOC); (**g**) dissolved organic carbon (DOC); (**h**) total nitrogen (TN); (**i**) ammonium nitrogen (NH_4_^+^); (**j**) nitrate nitrogen (NO_3_^−^); (**k**) dissolved organic nitrogen (DON); (**l**) total phosphorus (TP); (**m**) available phosphorus (AP); (**n**) total potassium (TK); and (**o**) available potassium (AK). Values are means ± standard errors (n = 4). CR, cropland rotation; PF, plantation forest; FG, forage grass; FF, mix of plantation forest and forage grass. Different lowercase letters suggest significant differences in different land-use types (*p* < 0.05).

**Figure 3 jof-10-00280-f003:**
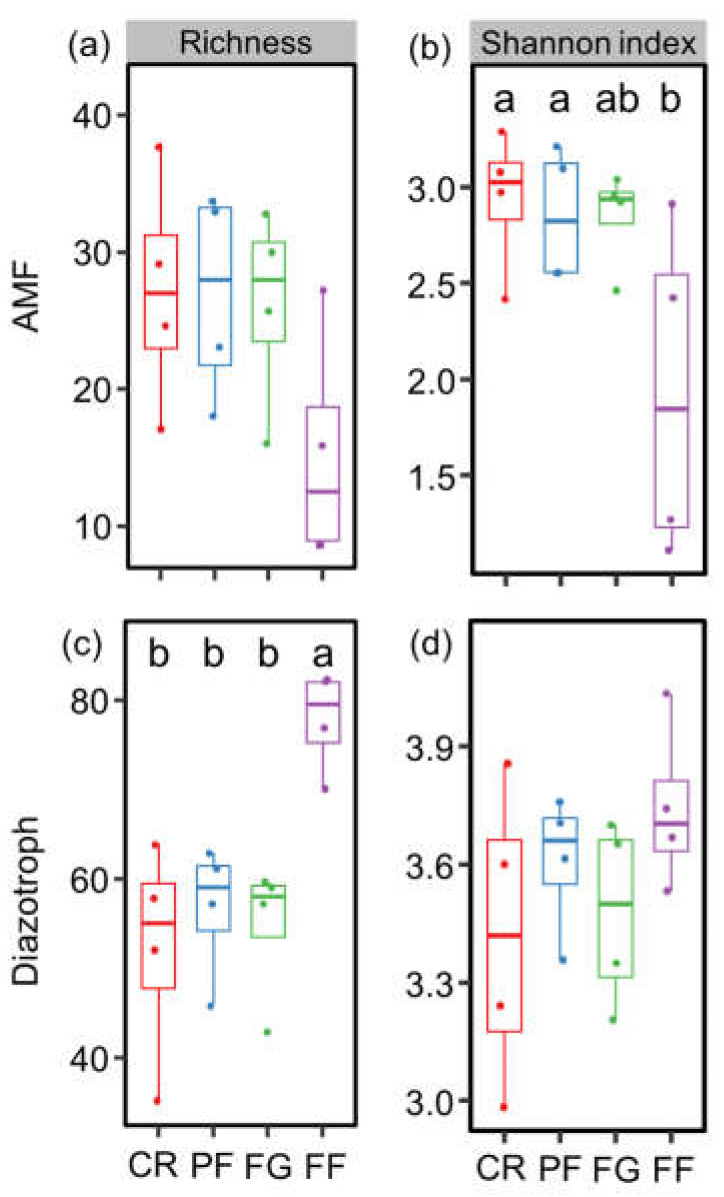
The diversity of (**a**,**b**) arbuscular mycorrhizal fungi (AMF) and (**c**,**d**) diazotrophs during vegetation restoration from cropland. Values are means ± standard errors (n = 4). CR, cropland rotation; PF, plantation forest; FG, forage grass; FF, mix of plantation forest and forage grass. Different lowercase letters suggest significant differences in different land-use types (*p* < 0.05).

**Figure 4 jof-10-00280-f004:**
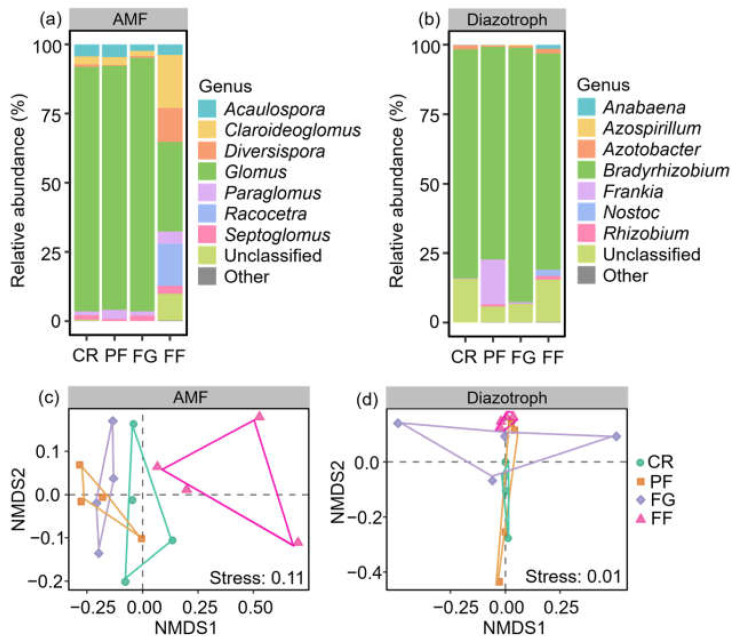
The differences in the community compositions of arbuscular mycorrhizal fungi (AMF) and diazotrophs during vegetation restoration from cropland. (**a**,**b**) Changes in the relative abundance of AMF and diazotrophs during vegetation restoration from cropland. (**c**,**d**) Non-metric multidimensional scaling (NMDS) of AMF and diazotrophs. CR, cropland rotation; PF, plantation forest; FG, forage grass; FF, mix of plantation forest and forage grass.

**Figure 5 jof-10-00280-f005:**
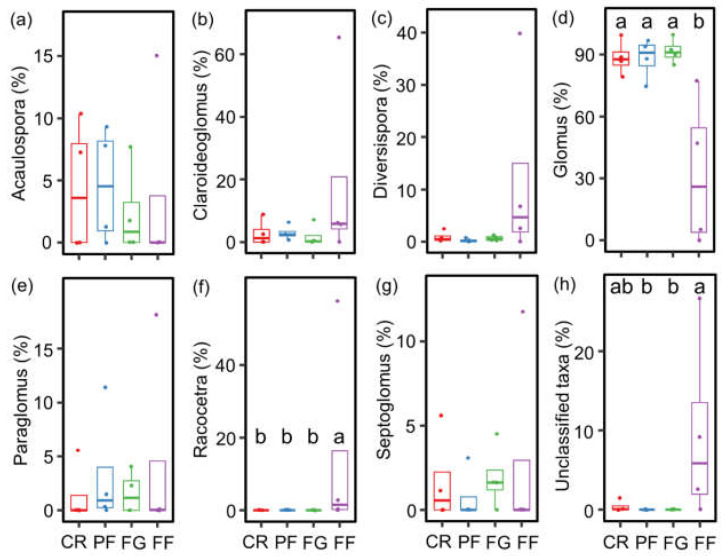
The relative abundance of arbuscular mycorrhizal fungi (AMF) at the genus level during vegetation restoration from cropland. (**a**) *Acaulospora*; (**b**) *Claroideoglomus*; (**c**) *Diversispora*; (**d**) *Glomus*; (**e**) *Paraglomus*; (**f**) *Racocetra*; (**g**) *Septoglomus*; and (**h**) unclassified taxa. Values are means ± standard errors (n = 4). CR, cropland rotation; PF, plantation forest; FG, forage grass; FF, mix of plantation forest and forage grass. Different lowercase letters suggest significant differences in different land-use types (*p* < 0.05).

**Figure 6 jof-10-00280-f006:**
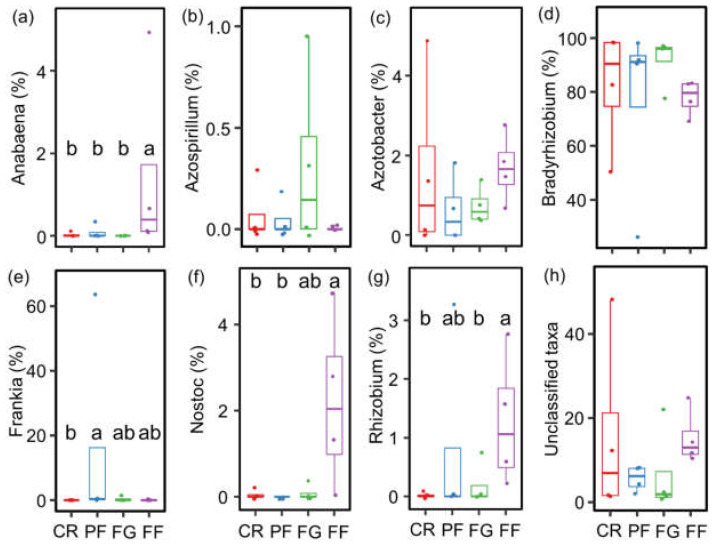
The relative abundance of diazotrophs at the genus level during vegetation restoration from cropland. (**a**) *Anabaena*; (**b**) *Azospirillum*; (**c**) *Azotobacter*; (**d**) *Bradyrhizobium*; (**e**) *Frankia*; (**f**) *Nostoc*; (**g**) *Rhizobium*; and (**h**) unclassified taxa. Values are means ± standard errors (n = 4). CR, cropland rotation; PF, plantation forest; FG, forage grass; FF, mix of plantation forest and forage grass. Different lowercase letters suggest significant differences in different land-use types (*p* < 0.05).

**Figure 7 jof-10-00280-f007:**
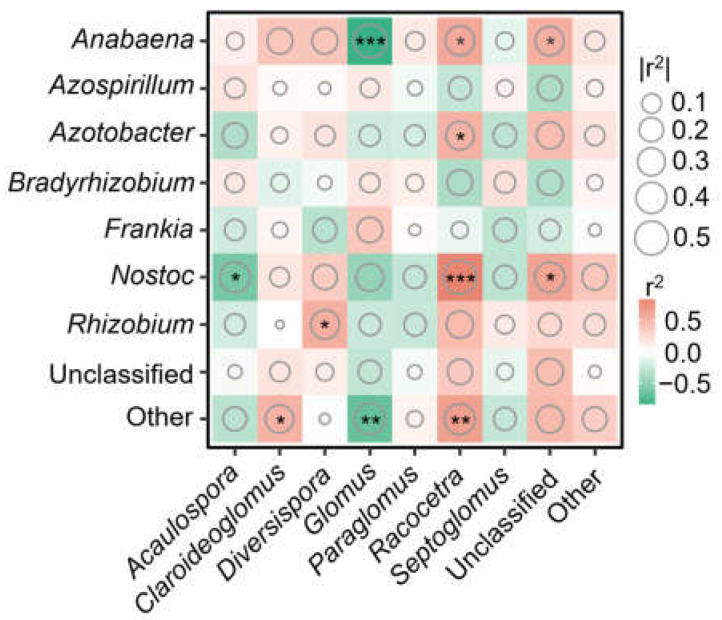
Pearson’s correlation between the genera of arbuscular mycorrhizal fungi (AMF) and diazotrophs. The “*” in the circle indicates significant correlations (* *p* < 0.05; ** *p* < 0.01; *** *p* < 0.001). The red and green suggest positive and negative correlation, respectively.

**Figure 8 jof-10-00280-f008:**
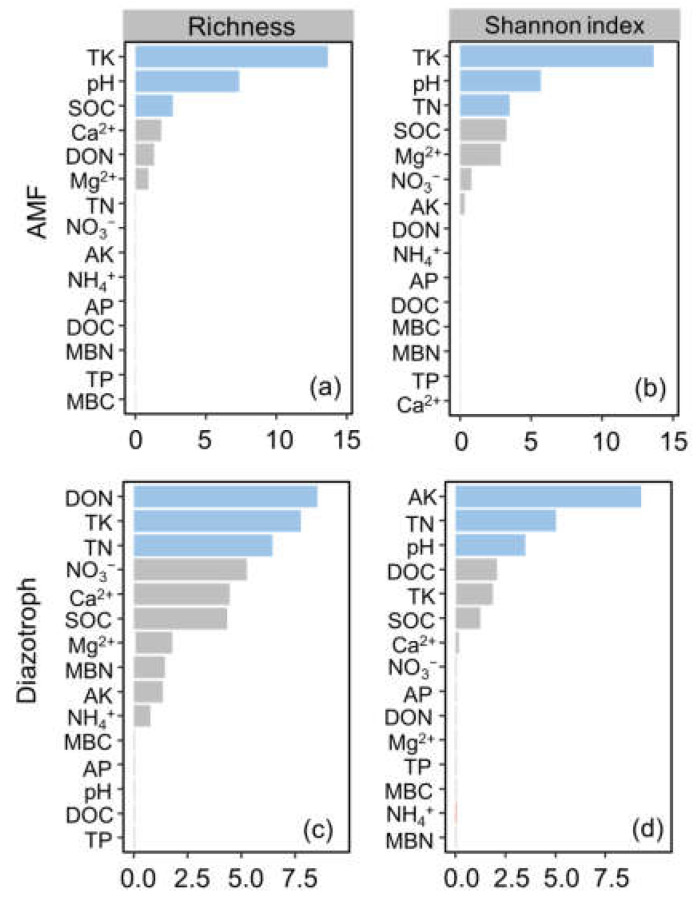
Relative importance of soil properties for an explanation of the variation with diversity of arbuscular mycorrhizal fungi (AMF) (**a**,**b**) and diazotrophs (**c**,**d**) based on the random forest models. Blue columns show the top three important factors. Ca^2+^ is soil exchangeable Ca^2+^; Mg^2+^ is soil exchangeable Mg^2+^; MBC is microbial biomass carbon; MBN is microbial biomass nitrogen; SOC is soil organic carbon; DOC is dissolved organic carbon; TN is total nitrogen; NH_4_^+^ is ammonium nitrogen; NO_3_^−^ is nitrate nitrogen; DON is dissolved organic nitrogen; TP is total phosphorus; AP is available phosphorus; TK is total potassium; AK is available potassium.

## Data Availability

The data used to support the findings of this study can be made available by the corresponding author upon request.

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
