# Peer review of "Impacts of Managed Vegetation Restoration on Arbuscular Mycorrhizal Fungi and Diazotrophs in Karst Ecosystems"

_jof, 2024, doi:10.3390/jof10040280_

Round 1

Reviewer 1 Report

In the present manuscript entitled “Impacts of Managed Vegetation Restoration on Arbuscular My- 2 corrhizal Fungi and Diazotrophs in Karst Ecosystems”, the authors performed an interesting study providing new insights  on how the vegetation restoration can influence, microbial ecosystem development. In this work, the authors studied and decribed the community structure and diversity of AMFs and diazotrophs in a karst region undergoing managed restoration of vegetation from cropland.

The experimentation was well performed and the methods of molecular biology adopted as PCR amplification, and high-throughput sequencing for microbial community determination as well as statistical analysis are reliable and effective.

The work is of theoretical and applied interest; the reliability of the results described by the authors in favor of FF versus PF and FG restoration strategies reduces any possibility of commentary.

The work is original and can be accepted if the authors respond to the following reviewer's comments

The authors in the manuscript correctly stated that soil properties such as total potassium, available potassium, pH and total nitrogen are to be considered as the main factors influencing AMF and diazotroph diversity.

I ask the authors why they did not also mention phosphate and ammonium content, which are considered limiting factors for AMF and diazotroph development? (see: Regulation of mycorrhiza development in durum wheat by P fertilization: Effect on plant nitrogen metabolism. DOI: 10.1002/jpln.201700110).

The authors among the diazotrophs point to Anabaena and Nostoc, which in addition to improving resistance to water erosion, provide available nutrients to plants through the gradual release of N and P to benefit soil fertility.

Unlike Rhizobium, which is a symbiotic nitrogen fixer and is typical of soil, Anabaena and Nostoc are cyanobacteria that require water and light for their photosynthetic activity. Can the authors better clarify for readers where cyanobacteria in the soil find suitable conditions for their development?

The authors' study is aimed at restoring vegetation in a karst area characterized by low-fertile soils and limited soil microflora to re-establish a proper ecosystem balance between plants, microorganisms and mineral nutrients;

Why did the authors not consider a fourth replanting strategy (using, based on the data obtained), sporigenic inocula of AMF and diazotrophs to promote plant vegetative recovery?

No detail comments.

Author Response

Comment 1: In the present manuscript entitled “Impacts of Managed Vegetation Restoration on Arbuscular My- 2 corrhizal Fungi and Diazotrophs in Karst Ecosystems”, the authors performed an interesting study providing new insights on how the vegetation restoration can influence, microbial ecosystem development. In this work, the authors studied and decribed the community structure and diversity of AMFs and diazotrophs in a karst region undergoing managed restoration of vegetation from cropland.

The experimentation was well performed and the methods of molecular biology adopted as PCR amplification, and high-throughput sequencing for microbial community determination as well as statistical analysis are reliable and effective.

The work is of theoretical and applied interest; the reliability of the results described by the authors in favor of FF versus PF and FG restoration strategies reduces any possibility of commentary.

The work is original and can be accepted if the authors respond to the following reviewer's comments

The authors in the manuscript correctly stated that soil properties such as total potassium, available potassium, pH and total nitrogen are to be considered as the main factors influencing AMF and diazotroph diversity.

Response 1: We thank Reviewer 1 for the helpful comments and thorough review. I have thoroughly revised the manuscript in accordance with the suggestions provided by the reviewer. The valuable feedback has been instrumental in enhancing the overall quality and clarity of the study. We appreciate the constructive comments and have addressed the questions and concerns raised during the review process. The revised manuscript now provides a more comprehensive and refined exploration of the effects of managed vegetation restoration strategies on the diversity and community composition of arbuscular mycorrhizal fungi (AMF) and diazotroph. I trust that the revisions will address the reviewer's concerns and contribute to the improved clarity and impact of the research. Thank you for the opportunity to revise and refine this study.

Comment 2: I ask the authors why they did not also mention phosphate and ammonium content, which are considered limiting factors for AMF and diazotroph development? (see: Regulation of mycorrhiza development in durum wheat by P fertilization: Effect on plant nitrogen metabolism. DOI: 10.1002/jpln.201700110).

Response 2: Thank you for the above suggestion. We strongly agree that soil nitrogen and phosphorus content is indeed key factors affecting the communities of AMF and diazotroph. We have determined the content of ammonium and available phosphorus in soil, and it has been found that they have a key effect on heavy azotobacter and arbuscular mycorrhizal fungi in previous study [31]. In order to be more convincing, we have added a comprehensive discussion in section 4.3 with reference to the literature you provided as follows: “One study also found the positive correlation between N availability and diazotrophs [31], which agrees with the findings of this study that soil N content (e.g., DON and TN) showed significant contributions to various with diazotroph diversity. Conversely, higher TP and AP contents were observed in the CR than in other restoration strategies. This suggests that excessive P availability may have an inhibitory effect on nitrogen-fixing microorganisms. Previous studies have shown that soil P levels are negatively correlated with N-fixing microorganisms and plant growth, as high P levels can limit plant growth and N assimilation [45].” Please see the revised manuscript lines 406-413. We hope this modification significantly improves this manuscript.

Reference:

  1. Xiao, D.; Hong, T.; Chen, M.; He, X.; Wang, K. Assessing the effect of slope position on the community assemblage of soil diazotrophs and root arbuscular mycorrhizal fungi. Fungi. 2023, 9(4).
  2. Di Martino, C.; Palumbo, G.; Vitullo, D.; Di Santo, P.; Fuggi, A. Regulation of mycorrhiza development in durum wheat by P fertilization: Effect on plant nitrogen metabolism. Plant Nutr. Soil Sci. 2018, 181:429-440.

Comment 3: The authors among the diazotrophs point to Anabaena and Nostoc, which in addition to improving resistance to water erosion, provide available nutrients to plants through the gradual release of N and P to benefit soil fertility.

Unlike Rhizobium, which is a symbiotic nitrogen fixer and is typical of soil, Anabaena and Nostoc are cyanobacteria that require water and light for their photosynthetic activity. Can the authors better clarify for readers where cyanobacteria in the soil find suitable conditions for their development?

Response 3: We thank the reviewer for highlighting this important consideration. We have added the following details to the revised manuscript: “In addition, low soil nutrient (e.g., NO3−, AP and AK) conditions may serve as the primary stimulus for the proliferation of cyanobacteria genera like Anabaena and Nostoc. These cyanobacteria have been observed to release nutrients, particularly N, thereby providing additional nutrients for plants. Additionally, Nostoc has been noted to thrive in barren soils, indicating its resilience in nutrient-poor environments [43, 44].” This addition will ensure that readers have a clearer understanding of the ecological niche and suitable conditions for cyanobacteria in managed vegetation restoration strategies. Please see revised manuscript lines 385-340.

Reference:

  1. Dodds, W.K.; Gudder, D.A.; Mollenhauer, D. The ecology of nostoc. J. Phycol. 1995, 31, 2–18.
  2. Li, J.; Xie, T.; Zhu, H.; Zhou, J.; Li, C.; Xiong, W.; Lin, X.; Wu, Y.; He, Z.; Li, X. Alkaline phosphatase activity mediates soil organic phosphorus mineralization in a subalpine forest ecosystem. Geoderma. 2021, 404:115376.

Comment 4: The authors' study is aimed at restoring vegetation in a karst area characterized by low-fertile soils and limited soil microflora to re-establish a proper ecosystem balance between plants, microorganisms and mineral nutrients;

Why did the authors not consider a fourth replanting strategy (using, based on the data obtained), sporigenic inocula of AMF and diazotrophs to promote plant vegetative recovery?

Response 4: Thank you for your valuable suggestions, which provide new ideas for our subsequent experiments. We will consider inoculating spores of AMF and diazotroph in restoration strategies of plantations that have not been destroyed to further detect the characteristics of AMF and diazotroph community in this treatment. Moreover, we have explained this point in the Discussion 4.3, please see revised manuscript lines 426-427. Thank you once again for your contribution.

Reviewer 2 Report

The article is relevant for the research domain of interaction between plants and soil microbial communities. 

Several minor corrections are required, please see the comments on the manuscript. 

The results section should be improved, please describe the results showed in Fig 8b

Please add details in Figures caption about all the abbreviations, please see the comments on the manuscript.

Please rewrite the conclusion in term of the significance of the results on the soil value and the effect of managed vegetation restoration. Please avoid the repetition of the results and discussions. 

Please review the English language.

Author Response

Major comments

The article is relevant for the research domain of interaction between plants and soil microbial communities.

Response: Thank you for your feedback and thorough review of the manuscript. We appreciate your attention to detail and your commitment to improving the quality of our manuscript. According to your nice suggestions, we have made extensive corrections to our previous draft. We have made the necessary revisions to enhance language and the readability of the manuscript. We have had the manuscript proofread by a native speaker before resubmission. We look forward to your further feedback and comments

Detail comments

(1) Several minor corrections are required, please see the comments on the manuscript.

Response1: We feel great thanks for your professional review work on our article. As you are concerned, there are several problems that need to be addressed. We have made complete revisions to your detailed comments in the manuscript and shown in red text. We appreciate you again for your thorough review to improve the quality of our manuscripts.

(2) The results section should be improved, please describe the results showed in Fig 8b

Response2: Thanks for your comments and apologize for the incomplete description of Figure 8. ln our resubmitted manuscript, we have added the details about Figure 8b as follows on: “Moreover, the variations observed in the Shannon index of AMF were mainly predicted by TK, pH, and TN (Figure 8b).” We believe that this information will provide the necessary clarity to our results regarding Figure 8b. Please see lines 310-312.

(3) Please add details in Figures caption about all the abbreviations, please see the comments on the manuscript.

Response3: We sincerely thank the reviewer for careful reading. As suggested by the reviewer, we have added details of all abbreviations in the Figures caption. Please see the revised manuscript.

(4) Please rewrite the conclusion in term of the significance of the results on the soil value and the effect of managed vegetation restoration. Please avoid the repetition of the results and discussions.

Response4: We appreciate your valuable comments. We have rewritten our conclusions according to your suggestion. Please see the revised manuscript lines 430-441. We look forward to your further positive feedback.

(5) Please review the English language.

Response5: Thank you for this comment. We have carefully reviewed the English language throughout the document to ensure clarity, coherence, and grammatical accuracy.

Reviewer 3 Report

The manuscript reports the response in the diversity of two important microbial communities to restoration strategies. The fact that it was performed in a inhospitable environment (degraded karst areas) make it relevant.

The presentation of the results is adequate. Some minor improvements in language are needed, and they are marked in the annotated file. 

The Discussion is well structured and written, except for Section 4.3, which needs improving, as the writing is truncated and the ideas are not clear. 

The Conclusion Section are a mixture of summary of results and repetition of some ideas presented in the Discussion. The authors must state clearly their main findings.

Author Response

Major comments

The manuscript reports the response in the diversity of two important microbial communities to restoration strategies. The fact that it was performed in a inhospitable environment (degraded karst areas) make it relevant.

Response: We would like to thank you for your careful reading, helpful comments, and constructive suggestions, which has significantly improved the presentation of our manuscript.

Detail comments

(1) The presentation of the results is adequate. Some minor improvements in language are needed, and they are marked in the annotated file.

Response1: Thank you for your feedback on the presentation of the results in our manuscript. We appreciate your thorough review and the comments highlighted in the annotated file. Your input is invaluable in refining the quality of our work, and we have adopted these improvements in full in our revised manuscripts. If you have any further comments or suggestions, please do not hesitate to let us know. Thank you for your time and attention to our manuscript.

(2) The Discussion is well structured and written, except for Section 4.3, which needs improving, as the writing is truncated and the ideas are not clear.

Response2: We thank the reviewer for reading our paper carefully and giving the above positive comments. We believe that the revised section 4.3 can clearly and accurately express the main views of this study. Please see the revised manuscript lines 403-427. We look forward to your further positive feedback.

(3) The Conclusion Section are a mixture of summary of results and repetition of some ideas presented in the Discussion. The authors must state clearly their main findings.

Response3: Thanks for your comments. We have completely rewritten the conclusions to avoid repetition with the results and discussions. In addition, we have shortened the paragraphs to make it easier for readers to identify our main findings. Please see the revised manuscript lines 430-441.

Round 2

Reviewer 1 Report

The authors have revised and improved, through the inclusion of new sentences and clarification of some concepts, as suggested in the reviewer's comments.

No detail comments